# Surgical Management, Prevention and Outcomes for Aneurysms of Arteriovenous Dialysis Fistulas: A Case Series Study and Review

**DOI:** 10.3390/ijerph20136256

**Published:** 2023-06-29

**Authors:** Adam Płoński, Adam Filip Płoński, Jerzy Głowiński

**Affiliations:** Department of Vascular Surgery and Transplantation, Medical University of Bialystok, 15-276 Bialystok, Poland; adamfplonski@yahoo.pl (A.F.P.); jglow@wp.pl (J.G.)

**Keywords:** arteriovenous fistula, dialysis, aneurysm, vascular access, Stentgraft, chronic kidney disease

## Abstract

The escalating population of individuals afflicted with end-stage renal disease necessitates the provision of secure and efficacious vascular access for hemodialysis, with arteriovenous fistulas representing a preferred option. Nonetheless, the creation of dialysis fistulas may entail complications, including the occurrence of fistula aneurysms that may require surgical intervention. This study included eight patients with large aneurysms of dialysis fistulas and aimed to evaluate the safety and effectiveness of classic, endovascular, or hybrid methods for fistula reconstruction or ligation, depending on the indications. Vascular interventions were performed on patients on chronic hemodialysis and on those in whom hemodialysis was discontinued due to the proper functioning of the transplanted kidney. Performed procedures were considered safe and effective. The reconstructed fistulas provided the patients with patent vascular access, allowing for continued hemodialysis. No re-aneurysmal dilatation of the reconstructed or ligated fistulas was observed. Regular monitoring of dialysis fistulas is crucial to detect complications in time. Guidelines should be established to specify the dimensions at which fistula aneurysm should be excised and whether to remove asymptomatic aneurysms at all. For patients who have undergone kidney transplantation, outlines should indicate when the fistula should be preserved and when it should be ligated.

## 1. Introduction

End-stage renal failure is a disease that can affect all organs. The increasing prevalence of affluent-society diseases such as obesity, diabetes, hypertension, or atherosclerosis means that the number of patients diagnosed with end-stage renal disease is systematically growing [1,2]. Stage G5 of chronic kidney disease, where GFR is <15 mL/min/1.73 m^2^, is the basic indication for renal replacement therapy. The recommended treatment in end-stage renal failure is kidney transplantation. However, the number of transplants performed in Poland has been very low for the last several years. In 2022, 857 kidneys were transplanted in Poland, including only 73 from living donors [3], while at the end of 2021 the number of dialysis patients was 19,416 [4]. Such figures are alarming, as many of those people are potential organ recipients. Due to the small number of kidney transplants, dialysis is the only alternative for renal replacement therapy in many patients with end-stage renal disease. The best vascular access for hemodialysis is a dialysis arteriovenous fistula [1,5,6,7]. Its benefits significantly exceed dialysis with central venous catheters [8,9]. The guidelines of the National Kidney Foundation Kidney Disease Outcomes Quality Initiative (K/DOQI) specify that the vascular access for hemodialysis characterized by the highest rate of long-term effectiveness is an arteriovenous fistula created from the patient’s own vessels [10]. These fistulas are preferred in patients who have no contraindications to their formation [1]. There is little data in the literature on the long-term complications of dialysis fistulas such as fistula aneurysm. This may be due to the fact that hemodialysis patients are often over 65 years old; due to their relatively short life expectancy, long-term follow-up is unknown. In addition, an aneurysm, if not inflamed, does not interfere with dialysis, and may not cause any symptoms. It is worth mentioning that in patients after kidney transplantation, the previously created fistula is relatively rarely closed [11]. The aim of this study was to present clinical cases of arteriovenous fistula aneurysms and the very diverse methods of their management, as well as to evaluate the effectiveness of the procedures performed.

## 2. Patients (Materials) and Methods

### 2.1. Patients

We conducted a review of clinical data on eight patients who underwent surgery for large dialysis fistula aneurysms at the Department of Vascular Surgery and Transplantation. The patients included seven males and one female, all of Polish nationality and Caucasian race, with ages ranging from 29 to 79 years old. Six of the patients had previously undergone kidney transplant and were on continuous immunosuppression, except for one patient who underwent graftectomy due to graft rejection. The patients were diagnosed with end-stage renal disease and underwent classic, endovascular, or hybrid methods for fistula reconstruction or ligation, depending on the indications.

### 2.2. Qualifications

The indications for dialysis fistula reconstruction were: progressive failure of the transplanted kidney requiring return to hemodialysis (applies to patients after kidney transplantation), graft rejection, impending thrombosis of the fistula, skin necrosis, bleeding, pain and inflammation, impending rupture, potential lack of available cannulation sites, and risk of infection. 

The indication for ligation of the fistula was no further need for hemodialysis in patients with a properly functioning transplanted kidney.

The qualification for fistula closure or reconstruction depended on several factors, including stability of the transplanted kidney function, the necessity to provide further hemodialysis, fistula hemodynamics (especially the presence of hyperkinetic flow), the coexistence of other fistula pathologies (stenosis or thrombus) increasing the risk of complications, and patients’ expectations and needs.

### 2.3. Fistula Examination

Before and after surgery, we performed Doppler ultrasound (DUS) examinations of the aneurysmal dilated dialysis fistulas using 4–11 MHz linear probes (Siemens ACUSON NX3 or Siemens ACUSON X300, Munich, Germany). We studied the fistula’s maximal diameter, aneurysm length, stenosis or thrombus presence, and blood flow characteristics. Based on the DUS examination, we assessed the possibility of fistula reconstruction or the necessity for ligation and the preferable type of surgery.

### 2.4. Surgical Procedure

The procedures were carried out by two experienced vascular surgeon consultants (AP, JG), each having conducted over 800 dialysis fistula operations. All eight patients underwent aneurysmectomy. For three of them, the fistula was ligated, and for five of them, the fistula was reconstructed. Four patients had the fistula reconstructed with a prosthesis, including one patient who had a stent graft BeGraft 6 × 39 (BeGraft peripheral, Bentley, Hechingen, Germany) implanted into the prosthesis. In one case, the fistula was reconstructed without the use of a prosthesis. In another case, a fragment of autogenous saphenous vein was used to reconstruct the brachial artery after aneurysm excision. For dialysis fistula reconstruction, if the anatomical conditions were favorable and there were indications that the fistula should be preserved, a reconstruction was performed after aneurysm excision using a new arteriovenous anastomosis, with a conversion from end-to-side to end-to-end anastomosis. This approach enabled the use of a proximal artery segment for reconstruction. However, if the anatomical conditions did not permit the reconstruction of the fistula using the remaining vessels, a prosthesis was used for reconstruction. The anastomoses between the prosthesis and the vessels were also end-to-end connections. In these cases, the fistula aneurysm was excised and replaced by a PTFE (W.L. Gore, Flagstaff, AZ, USA) prosthesis placed in subcutaneous channel. For reconstruction, we preferably used 8 mm standard wall PTFE but in one case, due to different dialysis fistula anatomical configurations, we used a 6 mm PTFE graft.

### 2.5. Follow-Up

After surgery, patients were routinely followed up and monitored according to the local protocol. Control clinical and DUS examinations were performed one month, three months, and six months after dialysis fistula reconstruction or ligation.

## 3. Results

### 3.1. Case Reports

#### 3.1.1. Case 1

A 45-year-old male patient with a history of end-stage renal failure in the course of polycystic kidney disease was admitted to the department. The patient had received a kidney transplant from a deceased donor 8 years earlier. He was admitted to the hospital due to the deterioration of the functioning parameters of the transplanted kidney in order to ameliorate the diagnosis and receive further treatment. The physical examination revealed the presence of an aneurysmal-dilated active arteriovenous fistula on the left forearm with a true aneurysm proximal to the arterio-venous anastomosis. The vascular access was made 9 years earlier—one year before the kidney transplantation. Diabetes mellitus, atherosclerosis, essential hypertension, and anemia were found among the comorbidities. The patient was on immunosuppressive therapy with prednisone and cyclosporine. A dialysis catheter was inserted into the patient’s internal jugular vein and the patient qualified for excision of the aneurysm. The fistula after kidney transplantation was not ligated. It was a radio-cephalic fistula from the patient’s own vessels in the area of the left wrist with end-to-side anastomosis of the vein with the artery. Due to the deteriorating functional parameters of the patient’s kidney, a decision was made to reconstruct the fistula. General anesthesia was applied. The skin was cut, the distal end of the artery was ligated, and the vein was clamped. A 4 cm × 3 cm aneurysm was excised entirely, within the limits of healthy vessels, and a new end-to-end anastomosis of the cephalic vein to the radial artery was made (Figure 1). A drain was left in the wound. The skin was sutured. The operation lasted 2 h and was uneventful. The patient did not require blood transfusions. The wound healed properly. A normal vascular murmur of the fistula was obtained. The patient was discharged home in good general condition.

#### 3.1.2. Case 2

A 38-year-old male patient with a history of end-stage renal failure was admitted to the department. He was admitted to the hospital because of a pulsating tumor in the area of the left arm that had been growing rapidly for several days, in order to ameliorate the diagnosis and receive further treatment. The patient had received a kidney transplant from a deceased donor 2 years earlier. The physical examination revealed the presence of an aneurysmal-dilated active arterio-venous fistula in the left arm with a true aneurysm proximal to the arterio-venous anastomosis. The area was inflamed, tense, and painful (Figure 2). Vascular access was created 6 months before kidney transplantation and was not ligated after it. Due to the pre-emptive transplantation, the patient was not included in the dialysis program and the fistula was never used. It was a brachio-cephalic fistula from the patient’s own vessels, in the area of the left arm with end-to-side anastomosis of the vein with the artery. Hypertension and anemia were found among the comorbidities. The patient was on immunosuppressive therapy with prednisone and cyclosporine. The patient qualified for excision of the aneurysm. Due to the rapid expansion of the aneurysm, the patient underwent emergency surgery. Due to the proper functioning parameters of the transplanted kidney, a decision was made to ligate the fistula and reconstruct the brachial artery. General anesthesia was applied. An incision was made in the upper half of the left arm and the brachial artery was dissected; the cephalic vein with a diameter of approx. 3 cm was dissected as well. The artery and vein were clamped. The skin over the aneurysm was cut. An aneurysm with a diameter of about 7 cm was dissected and excised entirely, within the limits of healthy vessels. The brachial artery was dissected in the area of anastomosis with the vein. The vein was cut off and the artery was sutured using a fragment of the left cephalic vein. The cephalic vein outflow of the fistula was ligated and stitched. A drain was left in the wound. The skin was sutured. The operation lasted 2 h and was uneventful. The patient did not require blood transfusions. The wound healed properly. The patient was discharged home in good general condition.

#### 3.1.3. Case 3

A 79-year-old male patient with a history of end-stage renal failure in the course of polycystic kidney disease was admitted to the department. He had received a kidney transplant from a deceased donor 13 years earlier. The patient had previously undergone a right-sided nephrectomy. He was admitted to the hospital due to intermittent bleeding from a throbbing and painful tumor on his left arm, in order to ameliorate the diagnosis and receive further treatment. The physical examination revealed the presence of an aneurysmal-dilated active arterio-venous fistula in the left arm with a true aneurysm proximal to the arterio-venous anastomosis. It was a brachio-cephalic fistula from the patient’s own vessels, in the area of the left arm with end-to-side anastomosis of the vein with the artery. The fistula was created 3 years before kidney transplantation and was not ligated after it. There was blood and purulent fluid oozing from the fistula. The area was inflamed, tense, and painful with visible partial tissue necrosis (Figure 3). The patient was on immunosuppressive therapy with prednisone and cyclosporine. He was diagnosed with hypertension and blood tests showed high leukocytosis and elevated inflammatory markers. The patient was qualified for excision of the aneurysm. Due to the symptoms of infection of the aneurysm, the patient underwent urgent surgery. Due to the correctly functioning parameters of the transplanted kidney, a decision was made to ligate the fistula. General anesthesia was applied. A teardrop incision was made above a large aneurysm of about 10 cm in diameter. The aneurysm was partially clotted with visible skin necrosis and purulent drainage. The aneurysm was dissected. An anastomosis with the brachial artery was dissected, ligated, punctured, and cut off. The outflow of the cephalic vein was dissected from the aneurysm, ligated, punctured, and cut off. An aneurysm measuring about 10 cm × 10 cm was excised entirely, within the limits of healthy vessels. The necrotic tissue over the fistula was excised as well. Skin plastic surgery was performed. A swab was taken from the ulcer above the fistula for bacteriological culture. *S. aureus* MSSA was cultured and antibiotic therapy with piperacillin and tazobactam was administered. Due to the large amount of tissue loss, a counter-incision was made. The wound was sutured with single sutures. After a week, a counter-incision was sutured as well. The operation itself was uneventful. The patient did not require blood transfusions. The wound healed properly. The patient was discharged home in good general condition.

#### 3.1.4. Case 4

A 79-year-old male patient with a history of end-stage renal failure was admitted to the department. The patient had been treated with hemodialysis for 7 years, firstly with the use of venous catheters, and currently for 3 years with a fistula on the right arm. He was admitted to the hospital due to rapidly pulsating and painful lumps in the area of his right arm, which had been growing for several days in order to ameliorate the diagnosis and receive further treatment. The area was tense and painful. The physical examination revealed the presence of an active arterio-venous fistula in the right arm with segmental true aneurysms along the course of the arterialized vein (Figure 4). It was a brachio-basilar fistula from the patient’s own vessels in the area of the right arm, created with the use of anterotransposition of the basilic vein, with end-to-side anastomosis of the vein with the artery. Hypertension and anemia were found among the comorbidities. The patient qualified for excision of the aneurysm. Due to the rapidly expanding aneurysm, the patient underwent urgent surgery. A decision was made to reconstruct the fistula. General anesthesia was applied. A lenticular incision was made in the upper half of the right arm and the anastomosis with the brachial artery and the basilic vein were dissected. The artery and vein were clamped. Two aneurysms with a diameter of approx. 5 cm and a dilated segment of the basilic vein with a length of approx. 15 cm were excised. In place of the removed vein, a PTFE Gore-Tex prosthesis with a diameter of 8 mm was sewn to the proximal and distal ends using end-to-end anastomoses. The operation lasted 1.5 h and was uneventful. The patient did not require blood transfusions. A normal vascular murmur of the fistula was obtained. The wound healed properly. The patient was discharged home in good general condition.

#### 3.1.5. Case 5

A 62-year-old male patient with a history of end-stage renal failure was admitted to the department. The patient had been treated with hemodialysis for 10 years, and currently for 2 years with a fistula on the left forearm. He was admitted to the hospital because of a rapidly pulsating and painful tumor in the area of the left forearm that had been growing for several days. The area was tense and painful. The physical examination revealed the presence of an aneurysmal-dilated active arterio-venous fistula in the left forearm with a true aneurysm proximal to the arterio-venous anastomosis (Figure 5). The fistula dated from 2 years earlier. It was a radio-cephalic fistula from the patient’s own vessels in the area of the left wrist with an end-to-side anastomosis of the vein with the artery. Comorbidities included hypertension and atherosclerosis. The patient qualified for excision of the aneurysm. A decision was made to reconstruct the fistula. Due to the rapidly expanding aneurysm, the patient underwent urgent surgery. General anesthesia was applied. The skin over the aneurysm was cut, the distal end of the artery was ligated, and the vein was clamped. An anastomosis of the artery was done and vein was dissected. An aneurysm measuring about 4 cm × 3 cm was excised entirely, within the limits of healthy vessels. A dilated fragment of the cephalic vein, approximately 10 cm long, was excised as well. In place of the removed vein, a PTFE Gore-Tex prosthesis with a diameter of 8 mm was sewn to the proximal and distal ends using end-to-end anastomoses. The operation lasted 1.5 h and was uneventful. The patient did not require blood transfusions. A normal vascular murmur of the fistula was obtained. The wound healed properly. The patient was discharged home in good general condition.

#### 3.1.6. Case 6

A 29-year-old male patient with a history of end-stage renal failure from early childhood was admitted to the department due to increasing hemodynamic disorders of a previously created fistula. The comorbidities included cataract, hyperparathyroidism, and atherosclerosis. The patient underwent bilateral nephrectomy at the age of three. After the procedure, he started peritoneal dialysis. At the age of nine, the patient developed his first hemodialysis fistula, a radio-cephalic fistula on the left forearm. After its maturation period, the patient began hemodialysis. At the age of 11, the patient underwent a kidney transplant procedure and the previously created fistula was ligated. The kidney functioned properly for 10 years. At the age of 21, the patient underwent a graftectomy due to the deteriorating functional parameters of the transplanted kidney and rejection of the transplant. In the same year, a decision was made to create a new hemodialysis access: a radio-cephalic fistula, on the right forearm. The fistula was active for two years, then it clotted. A decision was made to create another fistula, this time on the right arm. It was a brachio-cephalic fistula from the patient’s own vessels with an end-to-side anastomosis of the vein and artery. The fistula was punctured without complications for 6 years. After that time, it underwent aneurysmal dilatation. The patient qualified for excision of the aneurysm. Due to the need for further renal replacement therapy, a decision was made to reconstruct the fistula with a prosthesis. The patient was operated on as scheduled. A fragment of the dilated vein, about 8 cm in length, was removed and replaced with a Gore-Tex PTFE prosthesis with a diameter of 8 mm. After a month, thrombosis of the prosthesis was diagnosed and thrombectomy was performed. After 2 months, the patient was admitted to the department due to increasing hemodynamic disorders in the form of stenosis of the distal anastomosis of the PTFE prosthesis with the brachial artery. Ultrasound examination showed preserved flow through the fistula, with a slit-like flow lumen in the area of the anastomosis. The patient qualified for hybrid fistula reconstruction and was operated on under local anesthesia. The cephalic vein was punctured, angiography was performed, and stenosis in the anastomosis of the prosthesis with the brachial artery was visualized. A BeGraft Stentgraft with a diameter of 6 × 39 was implanted in the place of the stenosis, and post-dilation with a 7 mm diameter balloon was performed (Figure 6). The follow-up angiography showed a patent final image of the fistula. Clinically normal vascular murmur was obtained. The patient was discharged home in good general condition.

#### 3.1.7. Case 7

A 30-year-old female patient with a history of chronic renal failure from the age of 10 was admitted to the department. At the age of 15, she began hemodialysis with a brachio-cephalic fistula created from her own vessels using an end-to-side anastomosis of the vein with the artery in the left elbow fossa. At the age of 19 the patient underwent a kidney transplant. At the age of 24 the previously created fistula was ligated. The patient was admitted to the hospital due to a throbbing and painful lump in the area of the left cubital fossa, which had begun to enlarge after surgical ligation of the fistula performed 6 years earlier. Physical examination revealed a large pseudoaneurysm of a ligated arterio-venous fistula in the left arm, with local inflammation and skin necrosis. The arm was swollen, with a visible skin fistula. The patient was on immunosuppressive therapy with prednisone and cyclosporine. The patient was qualified for excision of the aneurysm and reconstruction of the brachial artery. Due to the expanding aneurysm, the patient underwent urgent surgery. General anesthesia was applied. An incision was made along the cubital fossa and the aneurysm was dissected. The artery and vein were clamped. The wall of the aneurysm, which was partially clotted, was incised. The thrombus was removed, and the liquefied content evacuated. The brachial artery was dissected. The tissues were covered with massive inflammatory infiltration. The tissues were cleaned and the aneurysm, with a diameter of approx. 6 cm, was excised. An arterial defect of approx. 4 cm in length was dissected. A decision was made to supplement the arterial defect with a fragment of the saphenous vein, which was taken from the patient’s left thigh. The continuity of the artery was restored with a vein. A drain was left in the wound and the skin was sutured. The blood supply to the hand was normal. The operation was uneventful. The patient did not require blood transfusions. The wound healed properly. The patient was discharged home in good general condition.

#### 3.1.8. Case 8

A 73-year-old male patient with a history of chronic renal failure was admitted to the department. He was admitted due to a rapidly enlarging, pulsating tumor in the area of the left arm in order to ameliorate the diagnosis and receive further treatment. The patient had received a kidney transplant from a deceased donor 8 years before. After 4 years, the kidney stopped functioning properly and the patient started dialysis through a brachio-cephalic fistula created 12 months earlier from his own vessels in the left elbow fossa. The physical examination revealed an aneurysmal-dilated, active arterio-venous fistula in the left arm with a true aneurysm proximal to the arterial-venous anastomosis. The area was inflamed, tense, and painful. The patient was on immunosuppressive therapy with prednisone and cyclosporine. The patient qualified for aneurysm excision and a decision was made to reconstruct the fistula. Due to the rapidly expanding aneurysm, the patient underwent emergency surgery. General anesthesia was applied. An incision was made over the aneurysm. The artery and vein were clamped. A dilated vein measuring approximately 5 cm × 8 cm was dissected and excised entirely. Collaterals were ligated in the proximal part. A large stenosis (up to 3 mm) was present at the outflow of the fistula, the probable cause of the aneurysm. The flow was restored by inserting a fragment of 6 mm Gore-Tex prosthesis, which was sewn to the vein cuff and to the cephalic vein. A drain was left in the wound. The skin was sutured. The operation lasted 1.5 h and was uneventful. The patient did not require blood transfusions. A normal vascular murmur of the fistula was obtained. The wound healed properly. The patient was discharged home in good general condition.

### 3.2. Summary

Each of the fistulas reconstructed by us enabled further hemodialysis. Obtained flow volume results are presented in the table according to follow up (Table 1). Surveillance protocol was accomplished at 1, 3, and 6 months post operation according to published recommendations [12,13]. During follow-up there was no hyperkinetic flow nor hemodynamically significant stenoses of the reconstructed fistulas detected, including the fistula in which we treated the previous stenosis by implanting a stent graft. In fistulas that were ligated, blood flow was not visualized. According to follow up, we did not record infection or potential graft rejection in any of the patients. Follow-up will be continued.

## 4. Discussion

The presence of a fistula aneurysm does not directly affect the course of dialysis and is not an absolute indication for discontinuation of its use [14]. However, such a large tumor is undoubtedly unsightly and poses a risk of hemorrhage when ruptured. Experience shows that even fistulas that have never been cannulated or have been cannulated a long time ago can undergo aneurysmal dilatation, as well [15]. In the case of patients on chronic hemodialysis, obtaining good vascular access is a key issue [1,5]. A dialysis fistula provides such access, but a vessel that is cannulated multiple times during dialysis may lose its mechanical strength. As a result of frequent puncturing of the fistula in the same place, the connective tissue fibers are destroyed. They have no chance to regenerate due to the short intervals between successive dialysis sessions. The vein wall becomes flaccid and tends towards aneurysmal dilatation [16]. High pressure in the fistula additionally predisposes it to this process [2]. The aneurysm grows according to LaPlace’s law. Turbulent blood flow through the fistula increases the value of shear forces acting in the vessel [17]. This may result in increased platelet aggregation and activation in the vessel wall and predispose it to vessel thrombosis. Maintaining the patency of the fistula and preventing its destruction may be facilitated by cannulating it along its entire length, each time in a different place. Dialysis station nurses and patients should be informed about this, as such conduct may result in long and proper functioning of the fistula. Active dialysis fistulas should be regularly monitored for early detection of complications of their use [1,18]. In the study conducted, out of 400 patients dialysed via an arterio-venous fistula, 129 developed fistula aneurysms [19]. These data emphasize the importance of regular monitoring of vascular access for maintaining a good quality of life in chronic hemodialysis patients. As the aneurysm widens, and the skin over it becomes more and more tense, it may become inflamed and painful. It has been proven that the majority of patients with dialysis fistula aneurysms have a significant degree of thinning and ulceration (82.5%) of the overlying skin [19]. The amount of tissue protecting the fistula is reduced, which increases the risk of rupture of the aneurysm. According to conducted meta-analysis, the most frequent indication for AVF aneurysm repair was bleeding prevention (86% of cases) [20]. In addition, local infectious complications pose a real risk of rupture [1,8]. Proceedings aimed at early management of dialysis fistula aneurysms before the occurrence of potential complications seem to be justified. In detecting potentially inefficient fistulas and estimating the complications of their use, color Doppler ultrasound may be a good choice. Moreover, clinical examination with a Doppler can be used as a surveillance protocol that can detect early AVF dysfunction at pair with fistulogram [21]. A separate issue is the decision to remove the fistula in patients with a properly functioning transplanted kidney [22]. In chronic arterio-venous fistulas, shear stress and high flow rates promote fibrosis of the vein wall and degeneration of the valvular apparatus. This causes the appearance of ectasia, a bending and coiling of the efferent vessel [17]. At the same time, high pressure in the fistula can weaken the vein wall for years and lead to its aneurysmal dilatation [2]. The expanding vessel causes a tumor effect; it can press on the surrounding nerves and cause pain. As our experience shows, this type of complication can occur even many years after the last fistula cannulation, and it can even occur in fistulas that have never been cannulated. It should be noted that in renal transplant patients, steroids taken as immunosuppressants may further increase the risk of aneurysmal vasodilation [2,11,23]. In chronic hemodialysis patients and, in particular, those who are immunosuppressed, it is quite important to try to avoid the use of dialysis catheters or AV grafts. According to a retrospective review comparing the results of aneurysmorrhaphy and graft interpositioning, interposition graft placement was associated with the loss of primary assisted patency, loss of secondary patency, and abandonment of dialysis access at 2 years. AVF aneurysmorrhaphy was associated with improved primary assisted and secondary patency and decreased abandonment of dialysis access [24]. A recently described technique involves creating a concomitant new simultaneous arterio-venous fistula in patients while continuing to use the primary failing aneurysmal AVF to avoid placement of an HDC or AVG. Once the new AVF becomes operational, the primary aneurysmal AVF can be abandoned and potentially closed [25]. However, undertaking this method requires careful cardiological observation in order to detect potential cardiac complications early on. In addition, long-term complications of arterio-venous fistula may include peripheral edema, heart failure, steal syndrome, pulmonary hypertension, and central venous dilatation. An issue that cannot be overlooked is the effect of high-flow shunt volume on heart efficiency. According to the study conducted, out of 214 chronic hemodialysis patients included in the analysis, heart failure was diagnosed in 57% of them [26]. This shows that a real threat for hemodialysis patients can be cardiac complications. Among patients with HF, HF with preserved ejection fraction (HFpEF) was, by far, the most common phenotype and occurred in 35% of them. Patients suffering from HFpEF were older and had not only typical echocardiographic changes but also higher hydration, which mirrored increased filling pressures of both ventricles, than patients without HF had [26]. These types of patients may be potential candidates for procedures aimed to reduce shunt blood flow and cardiac output while maintaining patent vascular access for dialysis. Proximal artery restriction combined with distal artery ligation may be a good solution for particularly high flow fistulas. In a retrospective analysis, patients treated with this method achieved a significant improvement in cardiac parameters [27]. Blood flow decreased from 2047.21 ± 398.08 mL/min to 1001.36 ± 240.42 mL/min, and blood flow/cardiac output decreased from 40.18% ± 6.76% to 22.34% ± 7.21%. Systolic pulmonary artery pressure decreased from 32.36 ± 8.56 mmHg to 27.57 ± 8.98 mmHg, indicating an improvement in right heart function. The dilemma is whether to decide to remove the unused fistula, which may expand in aneurysmal form and cause other previously mentioned complications, or not to remove the fistula in the case of deteriorating function parameters of the transplanted kidney [11]. There are no clear guidelines in this matter [2]. The maintenance of arterio-venous fistulas can potentially lead to various negative consequences, including adverse effects on cardiac function, aesthetic concerns, and complications such as bleeding and rupture. High-flow AVFs can exert excessive strain on the heart, which can induce or worsen pre-existing cardiac dysfunction. However, an independent enhancement in cardiac function is observed following kidney transplantation, regardless of the vascular access status [28]. In kidney transplant recipients, ligating high-flow AVFs not only improves cardiac function but also serves as a preventive measure against heart failure. For patients experiencing declining renal function and possessing high-flow AVFs, one potential option is the application of banding techniques to reduce the flow within the AVFs. Retaining AVFs after renal transplantation provides the advantage of immediate and optimal access in the event of transplant failure while potentially preserving kidney function. In the case of a decision to eliminate the fistula, it should be taken into account that re-creation of the fistula after ligation may be very difficult or even impossible. Perhaps there should be clear guidelines that will define the doctor’s procedure in such situations [2,16]. If the possibilities of creating a dialysis fistula on the upper limbs are exhausted, creating a fistula on the lower limb may be worth considering. However, such a procedure may cause complications in the form of critical lower limb ischaemia, which will require subsequent surgical interventions [29]. An important problem regarding aneurysms of arterio-venous fistulas is the lack of recommendations regarding the dimensions at which fistula aneurysm should be excised and whether to remove asymptomatic aneurysms at all. A reasonable alternative to total aneurysmectomy is partial aneurysmectomy. The short-term results of this type of procedure seem promising [30]. Nonetheless, it should be borne in mind that leaving a part of the weakened and potentially inflamed tissue as a functional element of the vessel may pose a certain risk of defined complications. The community of vascular surgeons strives to determine the methods of treatment applicable in unusual cases. A noteworthy method for large and especially tortuous, diffuse AVF aneurysms has been described [31]. In this procedure the luminal diameter is reduced, excess length is resected, and the new reconstructed AVF is re-tunneled for continued use. The primary disadvantage of this procedure is the extent of the dissection required, the need for a temporary tunneled catheter that must be removed later on, and especially long, longitudinal stitching line on the reconstructed fistula. Not all aneurysms we encounter are suitable for this procedure. It is important to acknowledge that fistula stenoses present a significant risk to their functionality, and even successful treatment of stenosis does not guarantee the absence of future recurrence. These observations are supported by both our clinical experience and the existing literature. A common practice in the treatment of this type of complication is the implantation of a stent. However, upper extremity complications secondary to these stents are less frequently discussed. A recently published case report describes a 43-year-old male with a right brachio-cephalic fistula who experienced symptoms of venous hypertension subsequent to the placement of a Wallstent for central venous stenosis [32]. Diagnostic evaluation revealed venous outflow obstruction resulting from stent foreshortening into the right subclavian vein. In situations where stent migration occurs and endovascular removal is not feasible, individual Wallstent fibers can be extracted through a limited venotomy. Modern techniques allow us to effectively supply the detected stenoses. However, the prevention and estimation of potential stenotic sites in a hemodynamically efficient fistula has its limitations. AVF stenosis may occur at sites with abnormal wall shear stress (WSS) and oscillatory shear index (OSI), which are caused by the complex flow in the AVF. A recently described ultrasound-based method allows us to determine the WSS and OSI values in different regions of the AVF to detect and analyze the risk sites [33]. According to the analysis, the possible risk site in the AVF may be located in the anastomosis and curved regions, where the latter could present a higher risk for AVF stenosis. The decisions made by the surgeons seem to have been the right ones. An end-to-end anastomosis of a vein to an artery has a lower risk of intimate hyperplasia than an end-to-side anastomosis. The applied anastomosis improves hemodynamic parameters and reduces the probability of vascular access stenosis [34,35,36]. Each of the presented cases was different and required the use of various surgical techniques. Decisions were often made intraoperatively and required the extensive experience of the operating team.

## 5. Conclusions

Dialysis fistulas should be regularly monitored for early detection of complications of their use. There should be clear guidelines specifying from what dimensions fistula aneurysm should be excised and whether to remove asymptomatic aneurysms at all. For patients after kidney transplantation, there should be outlines indicating when the fistula should be preserved and when it should be ligated.

## Figures and Tables

**Figure 1 ijerph-20-06256-f001:**
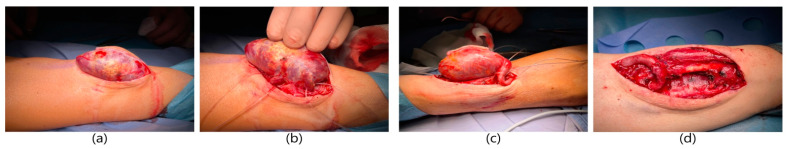
Surgical procedure of reconstruction of the radio-cephalic fistula. (**a**) Dissection of the fistula aneurysm. (**b**) Exposure of the radial arteries in the proximal and distal part of the fistula aneurysm. (**c**) Exposure of the cephalic vein (outflow of the fistula). (**d**) The new end-to-end anastomosis of the cephalic vein to the radial artery.

**Figure 2 ijerph-20-06256-f002:**
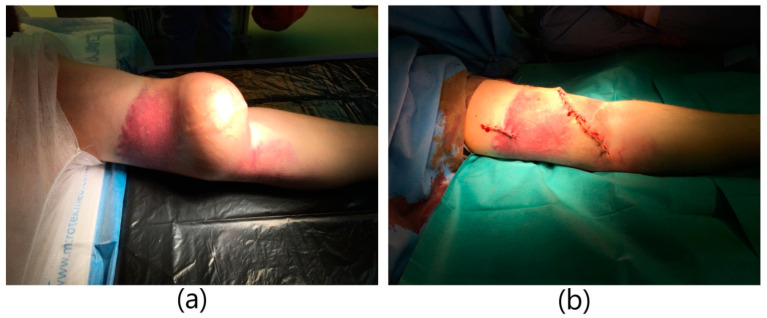
Huge aneurysm of brachio-cephalic fistula. (**a**) Aneurysm of the fistula with inflammatory infiltration of the skin. (**b**) Image immediately after excision of the aneurysm.

**Figure 3 ijerph-20-06256-f003:**
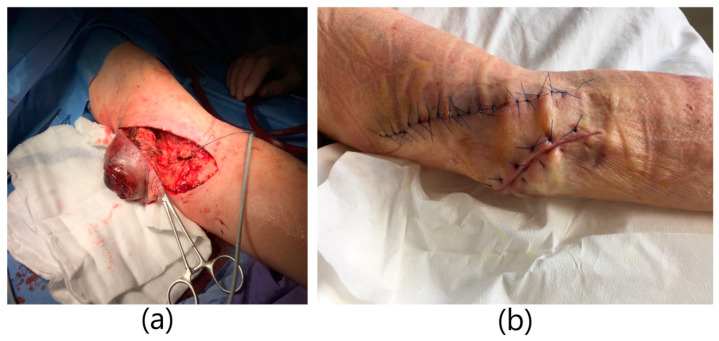
Infected brachio-cephalic fistula aneurysm with skin necrosis:. (**a**) Excision of the fistula aneurysm with skin changes. Ligation and cutting off the cephalic vein. (**b**) Image immediately after excision of the aneurysm.

**Figure 4 ijerph-20-06256-f004:**
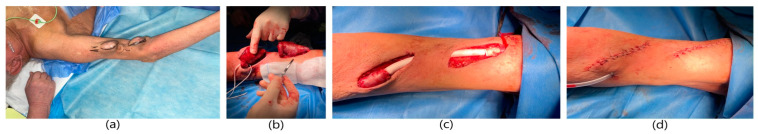
Surgical procedure of reconstruction of the brachio-basilar fistula aneurysm using PTFE prosthesis. (**a**) Aneurysms of the brachio-basilar fistula. (**b**) Dissection of the fistula’s aneurysms. Dissection and clamping of the basilic vein in the proximal and distal part of the aneurysms. (**c**) Implanted PTFE prosthesis in place of excised fistula aneurysms (end-to-end anastomoses). (**d**) Image immediately after excision of the aneurysm.

**Figure 5 ijerph-20-06256-f005:**
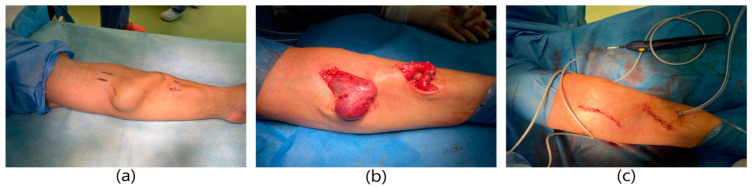
Surgical procedure of reconstruction the radio-cephalic fistula aneurysm using PTFE prosthesis. (**a**) Aneurysm of the radio-cephalic fistula aneurysm. (**b**) Dissection of the fistula aneurysm. Dissection of the cephalic vein in the proximal and distal part of the aneurysm. (**c**) Image immediately after excision of the aneurysm.

**Figure 6 ijerph-20-06256-f006:**
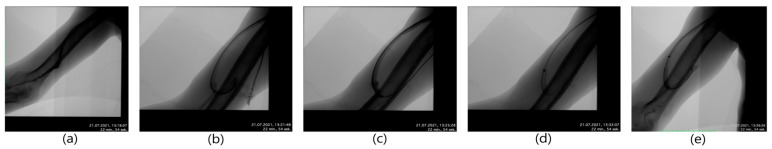
Implantation of a stent graft into a narrowed arterio-venous fistula. (**a**) Basilic vein cannulation. (**b**) Cannulation of the fistula and the brachial artery. (**c**) Implantation of the BeGraft stent graft in the place of the junction of the PTFE prosthesis with the brachial artery. (**d**) Terminal balloon angioplasty. (**e**) Final angiography.

**Table 1 ijerph-20-06256-t001:** Post operative blood flow volume in reconstructed fistulas among patients in the study group.

	Dialysis Fistula Flow Volume [mL/min]		
	1 Month	3 Months	6 Months
Patient 1	787	793	836
Patient 2	X	X	X
Patient 3	X	X	X
Patient 4	1328	1433	1472
Patient 5	1044	1086	1017
Patient 6	1004	1026	1045
Patient 7	X	X	X
Patient 8	1132	1227	1290

X—fistula ligated.

## Data Availability

The data presented in this study are available on request from the corresponding author. The data are not publicly available due to privacy.

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
