# Peer review of "Surgical Management, Prevention and Outcomes for Aneurysms of Arteriovenous Dialysis Fistulas: A Case Series Study and Review"

_ijerph, 2023, doi:10.3390/ijerph20136256_

Round 1
Reviewer 1 Report
The article describes in an excellent way the technical possibilities of surgical management of symptomatic dialysis shunt aneurysms. This is explained on 8 cases with corresponding visual material. These are often complex and lengthy procedures. The corresponding techniques have been known and established for many years. The article would therefore be ideally suited as a technical illustration and textbook.
In contrast, hardly any information is given on prevention and outcome after surgical correction. An ultrasound examination was performed at baseline, after one, three and 6 months according to the methodology section, but information on flow volumes and further pathologies such as stenoses are missing. Only in one case a recurrent stenosis after 2 months is described, which was then treated by stent graft. How about this data? Is there any information on the cause of aneurysm formation - prestenotic, postenotic?
The highly interesting effect of shunt flow volume on heart failure with the consequences of right heart failure after many years of high shunt flow volumes is also not discussed. Here, the aspect of when shunt flow reduction should be performed to prevent heart failure would be extremely interesting. Is cardiac echocardiography available in the patients described? If so, it would be good if the corresponding data were included.
It would also be essential to discuss when shunt vein aneurysms should be corrected preventively to avoid potential complications. It is quite important to avoid the need to replace the shunt vein with an arteriovenous graft. So, it would be better to perform aneurysmorrhapy before the shunt vein is no longer suitable for this. Especially in patients under immunosuppression, implantation of arteriovenous grafts always has a certain risk of infection. Do the authors have any data on this? Follow-up of 3 to 5 years would be useful to control for potential graft infection.
The question about the removal of dialysis shunts after successful transplantation is of course always to be discussed. In addition to renal function, flow volume and potential shunt complications, the possibilities of arteriovenous access in the future plays an important role. Can the authors provide information on their patients in this regard?
In summary, the focus of this paper is to describe technical options for surgical management of symptomatic dialysis shunt aneurysms. The highly interesting aspects of outcome and especially prevention of dialysis shunt and cardiac complications are not addressed.
Reviewer 2 Report
Dear editor,
I read with interest the manuscript written by Plonski et al., regarding the the clinical cases of 8 arterio-venous fistula aneurysms and the very diverse methods of their management as well as to evaluate the effectiveness of the procedures performed. The authors concluded that dialysis AVF should be regularly monitored for early detection of complications of their use and should be clear guidelines specifying from what dimensions fistula aneurysm should be excised and whether to remove asymptomatic aneurysms at all. Moreover, in case of patients after kidney transplantation, there should be outlines indicating when the fistula should be preserved and when it should be ligated.
Overall, the manuscript is well written and very well structured, and easy to read. Moreover, the quality of the Figure is decent.
However, I have some suggestion to improve the quality of the paper.
1. In Discussion Section, I suggest compare the management of the authors patients with other paper published recently in literature regarding the management of aneurysmal AVF. See the following article:
- https://www.mdpi.com/2075-1729/12/4/529
- https://doi.org/10.1111/j.1525-139X.2011.00990.x
- https://doi.org/10.1016/j.avsg.2016.08.046
- https://doi.org/10.1016/j.jvs.2009.10.122
- https://doi.org/10.1016/j.ejvs.2019.07.033
Round 2
Reviewer 1 Report
The paper has improved significantly as a result of the revision. Important aspects have been integrated.